# Using Size and Composition to Assess the Quality of Lunar Impact Glass Ages

**Pham Nguyen [1] and Nicolle Zellner [2],\***

[1]  Department of Physics and Astronomy, Michigan State University, East Lansing, MI 48824, USA; nguye258@msu.edu

[2]  Department of Physics, Albion College, Albion, MI 49224, USA

\*  Correspondence: nzellner@albion.edu; Tel.: +1-517-629-0465

**Abstract:** Determining the impact chronology of the Moon is an important yet challenging problem in planetary science even after decades of lunar samples and other analyses. In addition to crater counting statistics, orbital data, and dynamical models, well-constrained lunar sample ages are critical for proper interpretation of the Moon's impact chronology. To understand which properties of lunar impact glasses yield well-constrained ages, we evaluated the compositions and sizes of 119 Apollo 14, 15, 16, and 17 impact glass samples whose compositions and $^{40}Ar/^{39}Ar$ ages have already been published, and we present new data on 43 others. These additional data support previous findings that the composition and size of the glass are good indicators of the quality of the age plateau derived for each sample. We have further constrained those findings: Glasses of $\geq 200$ μm with a fraction of non-bridging oxygens (X(NBO)) of $\geq 0.23$ and a $K_2O$ (wt%) of $\geq 0.07$ are prime candidates for argon analyses and more likely to yield well-constrained $^{40}Ar/^{39}Ar$ ages. As a result, science resulting from impact glass analyses is maximized while analytical costs per glass are minimized. This has direct implications for future analyses of glass samples for both those in the current lunar collection and those that have yet to be collected.

**Keywords:** Apollo; lunar impact glass; $^{40}Ar/^{39}Ar$

## 1. Introduction

Lunar samples continue to be invaluable to the study of the Moon's impact history. The Moon's well-preserved surface and its close proximity allow for a better understanding of Earth's own impact history and can also help inform models of the solar system's dynamical evolution. Material recovered from the Apollo missions (e.g., bulk rock samples and regolith) along with lunar meteorites found on Earth provide important sources of information on these histories. Despite the wealth of data provided by these sources, however, the impact history of the Moon has proven to be complex and difficult to decipher [1,2]. Therefore, correct interpretation of the impact sample data is crucial for reconstructing an accurate impact history.

Impact glasses, which form from droplets of melt that have been ballistically transported from energetic impact events and are deposited into surrounding regolith, are one important source of data. These glasses retain the chemical composition of their target material and age of impact, making them powerful probes of regional geology and impact history [3]. However, interpretations must be made with caution. In order to make inferences of impact history, both the compositions and ages of glasses must be taken into account. For example, not all glasses are impact in origin. The Moon experienced a period of active volcanism between 3.8–3.2 Ga [4], so glasses must be screened for those that are volcanic in origin. Fortunately, the volcanic glasses can be characterized by a high $MgO/Al_2O_3$ ratio, chemical homogeneity, and Mg-correlated abundances of Ni, among other criteria (see [5] for

more details). Age data that are not further supplemented with geochemical data can also lead to overestimates in the number of impact events needed to explain glasses that have similar ages.

In this work, a set of 119 impact glasses whose compositions and age assessments have been characterized (e.g., [6–10]) from Apollo regolith samples 14259,624; 15221,21; 64501,225; 66041,127; and 71501,262, and new data from 43 others, were studied. We created bivariate density plots and applied statistical methods to quantify the factors—particularly size and composition—that are important for yielding well-constrained $^{40}$Ar/$^{39}$Ar ages.

## 2. Materials and Methods

The lithophile element compositions (e.g., $Al_2O_3$, MgO, and $TiO_2$) for the new set of 43 lunar impact glasses were determined by electron microprobe analysis, which allows for the determination of chemical composition in small samples and is well-suited for impact glasses that are typically hundreds of microns in size [5] (see Supplementary Table S1 for major-element data). Ages were determined using $^{40}$Ar/$^{39}$Ar dating by means of laser step-heating a glass sample to degas argon that is measured at each heating step. The $^{40}$Ar/$^{39}$Ar ratios represent an apparent age measurement that can be plotted against the amount of released $^{39}$Ar, creating an age spectrum (i.e., a "plateau", see Supplementry Figure S1). The age quality of a sample is determined by the number of steps that are statistically concordant (i.e., yield the same age within error), following the general practice of Jourdan (2012) [11]. Details of the geochemical and geochronological analyses can be found elsewhere [6–8].

Herein, $^{40}$Ar/$^{39}$Ar ages of impact glasses have been assessed according to how well-constrained each age is: Poor, fair, or good (as in [7]). Glasses with 'good' quality ages include >50% $^{39}$Ar in their age assessment [12] with most steps being concordant; 'fair' ages are similar but with fewer concordant steps; and 'poor' ages had no concordant steps. Age data of the 43 new glasses with 'good' and 'fair' age assessments can be found in Table 1 and "poor" glasses in Supplementary Table S2. Scatter plots and kernel density estimation plots (a type of probability distribution for bivariate data) were created to compare various glass characteristics (e.g., size, compositions, quality of age) and trends. Thus, by working backwards, i.e., determining the quality of $^{40}$Ar/$^{39}$Ar age for each glass and then determining what characteristics of the glass may have affected how well the age was measured, general guidelines for determining a priori which glasses (based on, for example, size and/or composition) are likely to yield well-constrained $^{40}$Ar/$^{39}$Ar ages have been established.

**Table 1.** K$_2$O (wt%), X(NBO), and age data for Apollo 14, 16, and 17 lunar impact glasses for good or fair glasses. "ND" means not determined. Assessment of argon release patterns are 'good' if >50% $^{39}$Ar was used in the age and most of the steps were concordant; 'fair' if some of the steps were concordant; and 'poor' if none of the steps were concordant.

| | Sample# | K2O (wt %) | X (NBO) | Age (Ma) | ± 2σ (Ma) | # Steps, % 39Ar Used in Age | Notes on Age | Assessment of Age | Size (μm) | Shape |
|---|---|---|---|---|---|---|---|---|---|---|
| Apollo 14 | 6 | 0.15 | 0.30 | 3733 | 592 | 2, 68.4 | weighted | fair | 150 | dumbbell |
| 14,259,624 | 8 | 0.26 | 0.30 | 825 | 126 | 9, 97.9 | weighted | good | 174 | shard |
| | 11 | 0.58 | 0.32 | 1310 | 20 | 6, 95 | plateau | good | 300 | shard |
| | 16 | 0.21 | 0.32 | 3557 | 249 | 8, 85 | plateau | good | 300 | shard |
| | 21 | 0.16 | 0.32 | 213 | 85 | 4, 97.8 | plateau | good | 250.5 | oblong |
| | 25 | 0.29 | 0.28 | 326 | 86 | 2, 100 | 2 steps | fair | 199.5 | dumbbell |
| | 26 | 0.50 | 0.33 | 1792 | 68 | 8, 91 | plateau | good | 300 | shard |
| | 29 | 0.43 | 0.31 | 1088 | 87 | 6, 73.6 | weighted | good | 250.5 | sphere |
| | 33 | 0.30 | 0.31 | 4442 | 429 | 3, 60 | weighted | fair | 174 | sphere |
| | 43 | 0.28 | 0.42 | 491 | 63 | 10, 93.4 | plateau | good | 250.5 | shard |
| | 47 | 0.88 | 0.32 | 3457 | 277 | 2, 97 | weighted | good | 150 | sphere |
| | 65 | 0.58 | 0.34 | 3798 | 226 | 1, 100 | 1 step | fair | 199.5 | shard |
| | 70 | 0.36 | 0.32 | 2709 | 388 | 1, 88 | 1 step | fair | 199.5 | sphere |
| | 123 | 0.14 | 0.29 | 2330 | 700 | 3, 100 | plateau | fair | 199.5 | sphere |
| | 125 | 0.61 | 0.29 | 3304 | 1636 | 2, 100 | weighted | fair | 199.5 | shard |
| | 145 | 0.40 | 0.29 | 2984 | 779 | 3, 90.8 | weighted | good | 150 | sphere |
| | 148 | 0.16 | 0.30 | 363 | 122 | 5, 100 | plateau | good | 250.5 | sphere |
| | 150 | 0.15 | 0.19 | 3610 | 2496 | 3, 100 | plateau | fair | 150 | dumbbell |
| | 158 | 0.72 | 0.21 | 2016 | 258 | 3, 75.6 | weighted | good | 199.5 | shard |
| | 160 | 0.35 | 0.36 | 3526 | 685 | 1, 79 | 1 step | fair | 199.5 | shard |
| | 163 | 0.45 | 0.30 | 3135 | 611 | 2, 94 | plateau | good | 199.5 | shard |
| | 167 | 0.47 | 0.28 | 106 | 19 | 6, 93.6 | plateau | good | 300 | shard |
| Apollo 16 | 191 | 0.11 | 0.32 | 1000 | 230 | 7, 67.3 | plateau | fair | 250.5 | shard |
| 64,501,225 | 204 | 0.21 | 0.24 | 3905 | 168 | 6, 90.7 | plateau | good | 349.5 | shard |
| | 207 | 0.10 | 0.36 | 925 | 358 | 8, 89 | plateau | good | 300 | shard |
| | 231 | 0.11 | 0.35 | 1573 | 190 | 6, 72.8 | plateau | good | 324 | shard |
| | 262 | 0.79 | 0.28 | 2818 | 249 | 2, 80.5 | weighted | fair | 199.5 | shard |
| Apollo 17 | 289 | 0.11 | 0.34 | 1323 | 904 | 4, 64 | weighted | fair | 199.5 | sphere |
| 71,501,262 | 375 | 0.33 | 0.26 | 3475 | 452 | 1, 100 | 1 step | fair | 250.5 | shard |

## 3. Results

In a series of heating experiments on large lunar glasses, Gombosi et al. (2015) determined that the fraction of non-bridging oxygens (X(NBO)) is inversely related to the diffusion of argon gas through glass [13]. In a companion study, Zellner and Delano (2015) [7] assessed hundreds of lunar impact glasses and proposed that glasses high in Fe and Ti were more likely to retain argon despite eons of diurnal heating of the lunar surface. Thus, glasses with larger X(NBO) should be more likely to retain argon [7,13–15]. Larger glasses, by virtue of their size, should also possess a larger quantity of argon than smaller ones.

Figure 1 shows a comparison between X(NBO) and the size of glasses of all age quality assessments. Spread exists across all age qualities making trend determinations somewhat difficult. To clearly determine whether or not any trends exist within or among the various data sets, glasses with 'poor' and 'good' age assessments were plotted separately and with a density estimate plot as shown in Figure 2.

The density plot estimates the distribution of size and X(NBO) data using concentric rings that fade to lighter colors with decreasing concentration of data. The density regions for glasses with age assessments qualified as 'good' are centered around larger sizes and higher X(NBO). These 'good' glasses have a median size of 260 μm while glasses with 'fair' age assessments have a median size of 208 μm. Based on the median values of size and X(NBO) of glasses with 'poor' age assessments, we estimate that a cutoff size of 200 μm and an X(NBO) of about 0.23 can be used to distinguish between 'poor' and 'good' quality age assessments (shown as dashed lines in Figure 2).

In addition to size and X(NBO), the amount of potassium in the glass is useful for distinguishing among glasses that are likely (or not) to yield well-constrained ages. Since $^{40}$K decays into $^{40}$Ar, it would be expected that samples with high $K_2O$ content would yield well-constrained ages. Indeed, it can be see that, in general, glasses with 'poor' and 'fair' age assessments have a lower $K_2O$ content (mean values of 0.196 and 0.237, respectively) compared to 'good' glasses (mean value 0.382), and we selected a cutoff of $K_2O$ (wt%) = 0.07, the median value of 'poor' glasses. There are exceptions, though, as some 'poor' glasses of size >200 μm have $K_2O$ (wt%) well in excess of the median value, so other criteria, such as in the value of X(NBO), should be considered during the selection process (Figure 3).

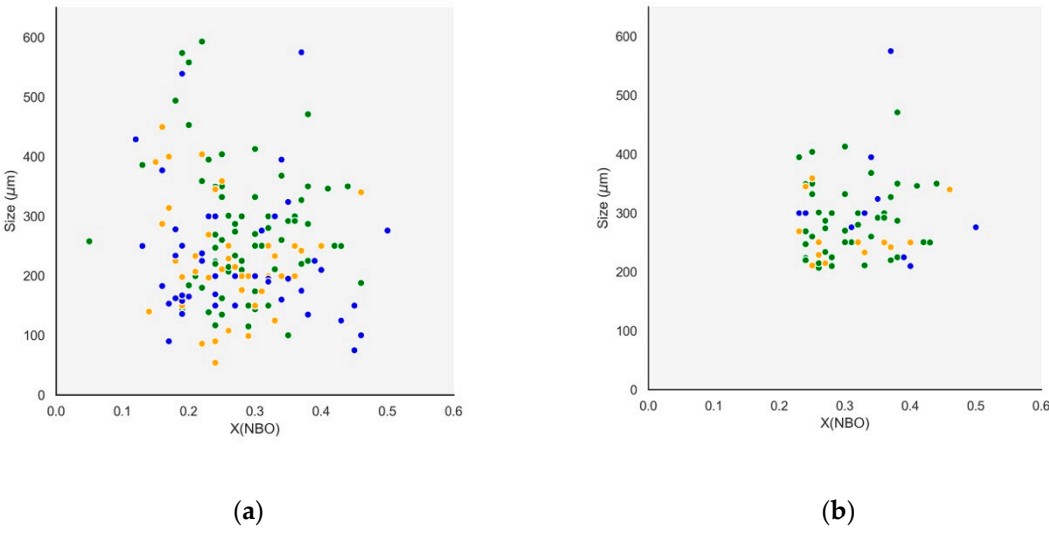

(**a**)　　　　　　　　　　　　　　　　　　　　　　　　　　　　　　(**b**)

**Figure 1.** (**a**) Plot comparing non-bridging oxygens (X(NBO)) and size for all 162 impact glasses with 'good' (green), 'fair' (yellow), and 'poor' (blue) age assessments. (**b**) Comparison of 74 glasses after imposing a minimum lower size limit of 200 μm and an X(NBO) ≥ 0.23. Data in both figures are from [6–10] and Table 1.

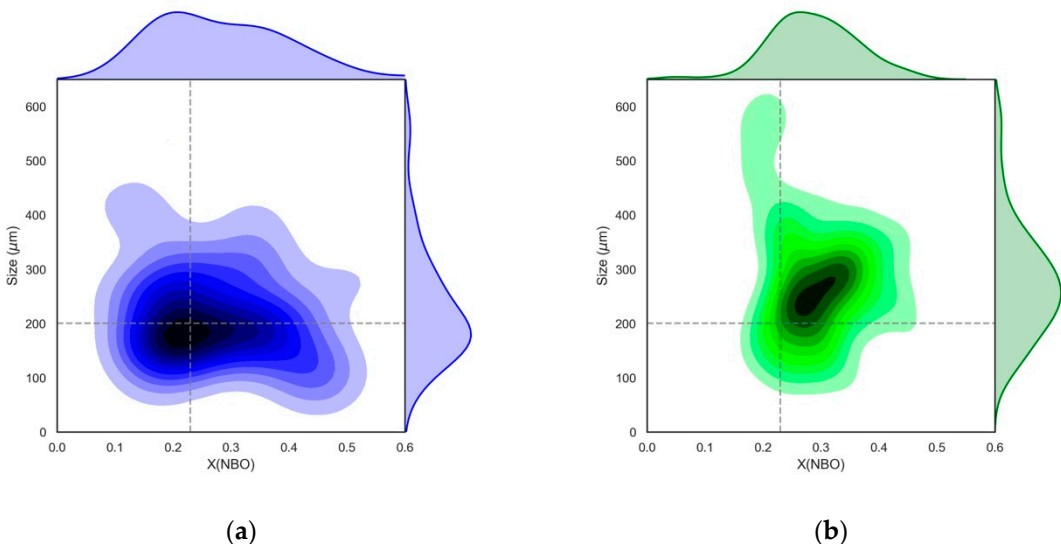

(**a**)                                                    (**b**)

**Figure 2.** Bivariate density plots of size and X(NBO) for (**a**) 'poor' glasses and (**b**) 'good' glasses (as defined in the text). Darker shading represents a higher spatial density of glasses for that region of the plot. Univariate distribution estimates for size and X(NBO) are shown on the right and top edges of each figure. Dashed lines indicate a median cutoff size of ≥200 μm and an X(NBO) value of ≥0.23. Data in both figures are from [6–10] and Table 1.

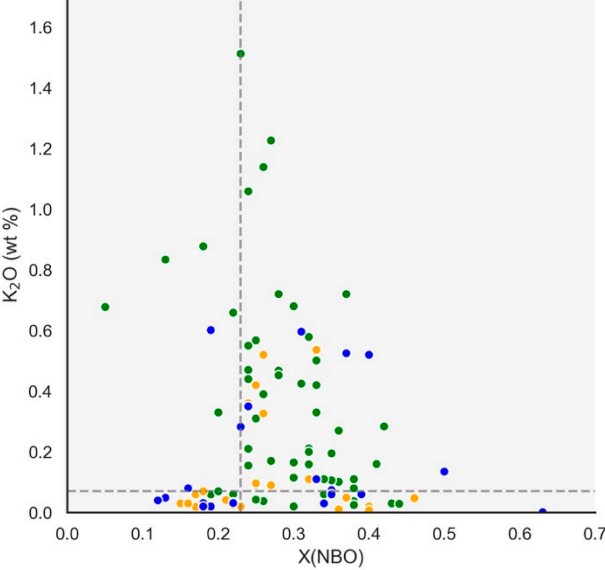

**Figure 3.** Plot comparing X(NBO) and $K_2O$ (wt%) for 100 impact glasses sizes of ≥200 μm with 'good' (green), 'fair' (yellow), and 'poor' (blue) age assessments (as defined in the text). $K_2O$ (wt%) = 0.07 and X(NBO) = 0.23 are indicated by the dashed lines. Data are from [6–10] and Table 1.

Table 2 shows how applying different criteria can affect the number of glasses of each age (quality) assessment compared to the fraction of 'good' glasses. Applying a single criterion such as size typically yields 51–56% of glasses having a 'good' age assessment, while using both size and X(NBO) increases the fraction of 'good' glasses by an additional 9–14%. With the additional consideration of $K_2O$ content, 71% of the glasses analyzed for $^{40}Ar/^{39}Ar$ yielded 'good' ages.

**Table 2.** Comparison of how applying various selection criteria to lunar impact glasses can affect the likelihood of analyzing glasses that will yield $^{40}\text{Ar}/^{39}\text{Ar}$ ages of 'good' quality. Data used from [6–10] and Table 1.

| Criteria | Fraction of Good Glasses | # Good | # Fair | # Poor | Total |
|---|---|---|---|---|---|
| Size Only ($\geq$200 μm) | 0.56 | 56 | 23 | 21 | 100 |
| X(NBO) Only ($\geq$0.23) | 0.51 | 61 | 31 | 28 | 120 |
| K$_2$O Only ($\geq$0.07) | 0.55 | 60 | 26 | 23 | 109 |
| Size and X(NBO) | 0.65 | 48 | 14 | 12 | 74 |
| Size, X(NBO), and K$_2$O | 0.71 | 39 | 8 | 8 | 55 |

## 4. Discussion

Zellner and Delano (2015) [7] derived a relationship between impact glass size and shape and argon gas diffusivity in impact glasses while studying a set of nearly 100 glasses from the Apollo 12, 14, 16, and 17 landing sites. The study determined that impact glasses with a X(NBO) of <0.25 were unlikely to yield reliable $^{40}\text{Ar}/^{39}\text{Ar}$ ages. Our expanded data set, which includes newly acquired data from 43 additional samples and samples from Apollo 15 [9], supports and further constrains the findings of the previous study. From this study, we find that along with limits on size and X(NBO), choosing glasses with K$_2$O (wt%) $\geq$ 0.07 will more likely result in well-constrained ages. It should also be noted that impact glass shape is another important selection factor. Prior work modeling the impact and transport of material on the Moon suggests that impact spheres are biased toward younger ages [16], as suggested by [7], while impact shards are more likely to be representative of total impact flux over all time [7].

The results of this study have direct implications for the future study of lunar samples. For example, three samples collected at the Apollo 15, 16, and 17 landing sites remain unopened, including two core vacuum sample containers (69001 and 73001) and one special environment sample container (15014). It has recently been suggested that one of these samples should be opened and analyzed [17]. Additionally, the proposed MoonRise and planned Chang'e 5 missions would return over a kilogram of rock fragments and regolith from the South Pole-Aitken Basin and two kilograms of regolith from the Mons Rumker region of Oceanus Procellarum, respectively. Analyses of these samples have the potential to further refine our understanding of the Moon's impact chronology, basin formation processes, and thermal evolution [18,19]. By using the criteria outlined in this paper, we can maximize the science potential of these samples.

## 5. Conclusions

The impact history of the Moon is complex and its proper interpretation requires careful analysis of sample data. In this study, we have presented data that support and further refine the parameters of size and composition to aid in and expedite the selection of choice lunar impact glasses for argon dating. In particular, by choosing glasses with a size $\geq$ 200 μm, X(NBO) $\geq$ 0.23, and K$_2$O $\geq$ 0.07, the maximum number of samples are most likely to yield well-constrained $^{40}\text{Ar}/^{39}\text{Ar}$ ages. These findings will be important to consider when selecting and studying lunar impact glasses that are likely to be brought to Earth during any future returns of lunar regolith samples, such as by the proposed Moonrise mission [18] and the planned Chang'e 5 mission [20].

**Supplementary Materials:** The following are available online at http://www.mdpi.com/2076-3263/9/2/85/s1, Table S1: Major-element compositions for Apollo 14, 16, and 17 lunar impact glasses, Table S2: K2O (wt%), X(NBO), and age data for Apollo 14, 16, and 17 lunar impact glasses, Figure S1: 40Ar/39Ar data for 43 newly analyzed Apollo 14, 16, and 17 lunar impact glasses.

**Author Contributions:** Data curation, N.Z.; formal analysis, P.N.; visualization, P.N.; writing—original draft preparation, N.Z. and P.N.; writing—review and editing, N.Z. and P.N.; supervision, N.Z.; funding acquisition, N.Z.

**Funding:** This research was funded by NSF Astronomy and Astrophysics grant #1516884 and by NASA Solar System Workings grant #NNX16AT34G.

**Acknowledgments:** NEBZ thanks Stefan Blachut, Oana Vesa, and Ray Cook for contributing to initial evaluations of trends in compositional data. These students were supported by Albion College's Foundation for Undergraduate Research, Scholarship, and Creative Activity. The authors thank Tim Swindle for his insight and guidance in interpreting $^{40}$Ar/$^{39}$Ar ages and Clark Isachsen and Sky Beard for assistance with obtaining the age data. The authors thank Vera Fernandes and two anonymous reviewers for helpful comments and suggestions that greatly improved the focus of this manuscript.

**Conflicts of Interest:** The authors declare no conflict of interest.

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
