# Peer review of "Using Size and Composition to Assess the Quality of Lunar Impact Glass Ages"

_geosciences, doi:10.3390/geosciences9020085_

Round 1

Reviewer 1 Report

See attached.

Author Response

Responses have been attached as a PDF file.

Reviewer 2 Report

Dear Authors,

It was we great delight I saw your manuscript.  I do think it's quite important and useful such work that overall helps out the community to improve its approach to data collection and interpretation.

Please find along the attached annotated pdf (sorry, it would be easier in word, I know) my edits, comments and suggestions to further improve your paper and make it an important help-guide for all of us.  Especially, there are a few places I request for more information to be provided instead of referring the reader to a citation. I believe that if you desire your contribution to be a guideline, it would be preferable to have the essential information in it.  These requests also come as a result of my not being completely familiar with certain terminology and its use, which perhaps is similar to others.

Please do not hesitate in further contacting me in case anything is unclear.

Wishing you a happy Festive season and a Great 2019.

Author Response

(The authors gave the same response as above.)

Reviewer 3 Report

The results of this manuscript are not surprising, but it quantifies some important factors very nicely. I think that it is very much worth publication, and I believe it requires only minor revision.

My most significant suggestion is to make a couple of the comparisons more quantitative, similar to what was done for size and X(NBO). What is the mean potassium content for good, fair, and poor analyses? Similarly, it would make sense to quantitatively compare the spheres and shards in Fig. 4, via medians ages of the two types, fractions <1 Ga, fractions >3 Ga, and/or some other quantitative measure.

Also, several of the references are in formats that are either inconsistent or misleading.

              #2, “Planetary Materials” is Vol. 36 of Reviews in Mineralogy and Geochemistry. It should be referenced by the journal or as a book with editors, not a mix.

              #7, the publisher of the book should be given.

              #10, #11, and #16 are abstracts, and I suspect #17 is as well. They should be referenced as “Abstract No.” The Lunar and Planetary Institute suggests a preferred referencing format, which should probably be used (with adjustments for the journal’s style).

              #13 should include the journal name, not “/gca”

              #14 is incomplete

I have a few other minor comments, given by line number:         

31: It might be good to include references to some papers about this confusion, such as the review by Bottke and Norman (2017, Ann. Rev. EPS 45, 619)

67: “microprobe”, not “microbe”

70: It’s actually an “apparent” age measurement, not an age measurement per se.

76-77: Jourdan would say it takes 70% to be “good,” but see Swindle & Weirich, LPSC 48, #1265.

Author Response

(The authors gave the same response as above.)

Round 2

Reviewer 1 Report

The paper is improved. There has been a change of phrasing that might not be the best choice. Regarding age assessments, the new phrase "well-resolved" is not the best. "Resolve" usually implies the ability to just separate two closely measured things. In this context, the authors original phrasing of "well-constrained" is MUCH more appropriate for the current paper.
Table 1: Title should include phrasing to tell reader this data is not the complete set of new data. Perhaps something like "...for good or fair glasses"?
Line 101: What does "Comparison of X glasses..." mean?
Fig 3: Why not put a dashed line for the (somewhat) arbitrary X(NBO) cut-off at 0.23?
Table S1 units: Can't just say Ox%. Must specify Wt. % or mol % or whatever the correct unit is.

Author Response

See attached PDF for author responses.
